# An Information Entropy-Based Modeling Method for the Measurement System

**DOI:** 10.3390/e21070691

**Published:** 2019-07-15

**Authors:** Li Kong, Hao Pan, Xuewei Li, Shuangbao Ma, Qi Xu, Kaibo Zhou

**Affiliations:** 1School of Artificial Intelligence and Automation, Key Laboratory of Image Processing and Intelligent Control, Ministry of Education, Huazhong University of Science and Technology, Wuhan 430074, China; 2School of Intelligent Engineering, Henan Institute of Technology, Xinxiang 453003, China; 3School of Mechanical Engineering and Automation, Wuhan Textile University, Wuhan 430074, China

**Keywords:** information entropy, modeling, measurement system, uncertainty, information acquisition

## Abstract

Measurement is a key method to obtain information from the real world and is widely used in human life. A unified model of measurement systems is critical to the design and optimization of measurement systems. However, the existing models of measurement systems are too abstract. To a certain extent, this makes it difficult to have a clear overall understanding of measurement systems and how to implement information acquisition. Meanwhile, this also leads to limitations in the application of these models. Information entropy is a measure of information or uncertainty of a random variable and has strong representation ability. In this paper, an information entropy-based modeling method for measurement system is proposed. First, a modeling idea based on the viewpoint of information and uncertainty is described. Second, an entropy balance equation based on the chain rule for entropy is proposed for system modeling. Then, the entropy balance equation is used to establish the information entropy-based model of the measurement system. Finally, three cases of typical measurement units or processes are analyzed using the proposed method. Compared with the existing modeling approaches, the proposed method considers the modeling problem from the perspective of information and uncertainty. It focuses on the information loss of the measurand in the transmission process and the characterization of the specific role of the measurement unit. The proposed model can intuitively describe the processing and changes of information in the measurement system. It does not conflict with the existing models of the measurement system, but can complement the existing models of measurement systems, thus further enriching the existing measurement theory.

## 1. Introduction

Measurement has been developed through the physical sciences and plays a very important role in industry, commerce, health and safety, and environmental protection [1,2,3,4,5]. A unified model of measurement systems is critical to the design and optimization of measurement systems. However, the existing measurement theory which will be reviewed below is too abstract. To a certain extent, this makes it difficult to have a clear overall understanding of measurement systems and how to obtain information with measurement units during the measurement process at the outset. Therefore, measurement science needs a theoretical framework [2] that can intuitively describe, analyze, and evaluate measurement systems and characterize how measurement units work to obtain the information of the measurand.

Numerous works of the modeling of the measurement or measurement system have been developed and published. Helmholtz and Hoelder developed a theory of measurement based on the concepts of the physical sciences [1], which regarded measurement as the operation set of assigning the determinate numerical value to the physical quantity of the object [2]. Then, three main model approaches or theories, the representational theory, the object-oriented method, and the probabilistic theory [6,7,8,9,10,11,12,13,14,15,16,17,18,19,20,21,22,23,24,25] were developed and studied. As the main body of these studies, the representational theory [6,7,8,9,10,11,12,13,14,15,16] considers representing the mapping between the measurand and measurement result with general symbols, from different standpoints such as semiotics [7,9], set theory [13], or domain-theory [15]. The object-oriented method [17,18,19] applies the object-oriented technology in computer programming to construct an object-oriented model of the measurement system, which divides the measurement elements into five classes described by their attributes, operations, or environment. The probabilistic theory [20,21,22,23,24,25] proposes a complete theory of measurement, including probability representations of different measurement scales, probabilistic descriptions of measurement systems, and measurement processes.

The abovementioned researches proposed the modeling methods from different perspectives or according to relevant theories. The models of measurement, measurement processes, or measurement systems were established, and even the special measurement problems, like in ratio, interval, ordinal, and nominal scales were adequately considered [13,16,21,22,25]. However, the implementation of the measurement relies on a series of measurement units. These theoretical models have considered the measurability, the relationship between the input and output of the system, calibration, restitution, etc., but cannot describe the role of the measurement unit in the measurement process. Furthermore, one of the cores of all measurements is the problem of uncertainty [26,27], but a model of measurement systems established directly from uncertainty has not been proposed.

Since absolute zero is impossible to achieve and the measured object always interacts with the outside world, the absolute standard measurement environment is not possible. This causes the measurand to be essentially a random process or a random sequence [28]. If the measurand is stationary, it can be considered as a random variable within a short measurement time. After the effective measurement, the uncertainty of the measurand is reduced compared to the uncertainty of the measurand before measurement. Therefore, the measurement is a process of uncertainty reduction, and its essence is information acquisition. Additionally, with the development of measurement science, the viewpoint which views measurement as an information process and regards instruments as information machines is widely recognized [1,2,9,29,30]. Since Shannon proposed the concept of information entropy as a measure of information and uncertainty of a random variable [31], information theory has been applied to some aspects of measurement [32,33,34]. Therefore, information entropy has high potential as one of the methods to solve measurement problems and it is feasible to establish a system model from the perspective of uncertainty with information entropy.

In this paper, an information entropy-based modeling method for measurement system is proposed. The main contributions of this paper are as follows: (1) a modeling idea based on the viewpoint of information and uncertainty is presented; (2) the entropy balance equation based on the chain rule for entropy is proposed for system modeling; and (3) an information entropy-based model of measurement systems is established based on the entropy balance equation from the perspective of uncertainty and information acquisition.

The rest of this paper is organized as follows. Section 2 presents the preliminaries on relations between different entropies, and proposes the entropy balance equation. The information entropy-based model of the measurement system is proposed in Section 3. Section 4 analyzes three cases of typical measurement units or processes with the proposed method. Finally, conclusions of the proposed model of the measurement system are drawn in Section 5.

## 2. Methodology

### 2.1. Information Entropy and Related Concepts

Entropy is a measure of the uncertainty of a random variable. For a discrete random variable X with limited states, probability of each state $X=x_{i}$, i=1,2,⋯,N, is denoted as $p\left(x_{i}\right)=p\left(X=x_{i}\right)$. For the sake of simplicity, we use $p\left(x_{i}\right)$ to represent probability instead of $p\left(X=x_{i}\right)$. Similarly, for discrete random variable Y, its probability function is denoted as $p\left(y_{j}\right)$, j=1,2,⋯,M. The joint probability function of X and Y is represented by $p\left(x_{i}y_{j}\right)$.

**Definition** **1.***The information entropy of the discrete random variable*X*is defined as*:(1)$$H\left(X\right)=-\sum_{i=1}^{N}p\left(x_{i}\right)logp\left(x_{i}\right).$$

If the log is to base 2, the unit of information entropy is bits; if the log is to base e (the natural logarithm), the unit is nats, and if the log is to base 10, the unit is harts. For the related measures that will be introduced later, their units are the same. The unit of entropy and related measures for continuous random variables is also the same.

For a continuous random variable X with probability density function p(x), its information entropy is infinite since the number of its stats is infinite. In this case, information entropy is the sum of differential entropy and a constant that tends to infinity. The definition of differential entropy is given as follows:

**Definition** **2.***The differential entropy of the continuous random variable X with probability density function*p(x), x∈R*, is defined as*:(2)$$h\left(X\right)=-\int_{R}p\left(x\right)logp\left(x\right)dx.$$

Obviously, differential entropy cannot represent the uncertainty of continuous random variables and does not have the connotation of information. However, when discussing mutual information, since two infinite constant terms will cancel each other, differential entropy has the same information characteristics as information entropy.

In this paper, in order to make each item in the model established in Section 3 have the connotation of information, the uncertainty of a random variable is characterized by information entropy, whether the random variable is continuous or discrete. In addition, for continuous cases, mutual information is calculated using differential entropy.

Based on the information entropy, the related concepts and their definitions are introduced below:

**Definition** **3.***The joint information entropy of discrete random variables*X*and*Y*is defined as*:(3)$$H\left(X,Y\right)=-\sum_{i=1}^{N}\sum_{j=1}^{M}p\left(x_{i}y_{j}\right)\log p\left(x_{i}y_{j}\right).$$

**Definition** **4.***The conditional entropy of the discrete random variable*X*given*Y*is defined as*:(4)$$H\left(X|Y\right)=-\sum_{i=1}^{N}\sum_{j=1}^{M}p\left(x_{i}y_{j}\right)\log p\left(x_{i}|y_{j}\right).$$

**Definition** **5.***The average mutual information (also referred to as mutual information) between discrete random variables*X*and*Y*is defined as*:(5)$$I\left(X;Y\right)=\sum_{i=1}^{N}\sum_{j=1}^{M}p\left(x_{i}y_{j}\right)\log\frac{p\left(x_{i}y_{j}\right)}{p\left(x_{i}\right)p\left(y_{j}\right)}.$$

The relationship between H(X), H(Y), H(X|Y), H(Y|X) and I(X;Y) can be expressed by the Venn diagram shown in Figure 1. Two equations governing this are:(6)$$\begin{array}{ll} H\left(X,Y\right) & =H\left(X\right)+H\left(Y|X\right)=H\left(Y\right)+H\left(X|Y\right)\\ & =H\left(X|Y\right)+H\left(Y|X\right)+I\left(X;Y\right) \end{array},$$
(7)$$\begin{array}{ll} I\left(X;Y\right) & =H\left(X\right)-H\left(X|Y\right)=H\left(Y\right)-H\left(Y|X\right)\\ & =H\left(X\right)+H\left(Y\right)-H\left(X,Y\right) \end{array}.$$

### 2.2. Entropy Balance Equation

In this part, the extension for the chain rule of joint entropy, called the entropy balance equation (Equation (8)), is developed for system modeling, which is given and proved below:

**Theorem** **1.***Given random variables*$X_{1},X_{2},\cdots ,X_{n}$*which are drawn according to*$p\left(x_{1},x_{2},\cdots ,x_{n}\right)$*, then*:(8)$$H\left(X_{1}\right)+\sum_{i=1}^{n-1}H\left(X_{i+1}|X_{i}X_{i-1}\cdots X_{1}\right)=H\left(X_{n}\right)+\sum_{i=2}^{n}H\left(X_{i-1}|X_{i}X_{i+1}\cdots X_{n}\right).$$

**Proof.** By the chain rule for entropy [35], we have:
(9)$$H\left(X_{1},X_{2},\cdots ,X_{n}\right)=\sum_{i=1}^{n}H\left(X_{i}|X_{i-1},X_{i-2},\cdots X_{1}\right)$$Equation (9) can be readily proved with $p\left(x_{1},x_{2},\cdots ,x_{n}\right)=\prod_{i=1}^{n}p\left(x_{i}|x_{i-1},x_{i-2},\cdots ,x_{1}\right)$ and the definitions of entropy and conditional entropy. By symmetry, one can write:(10)$$p\left(x_{1},x_{2},\cdots ,x_{n}\right)=\prod_{i=1}^{n}p\left(x_{i}|x_{i+1},x_{i+2},\cdots ,x_{n}\right)$$Thus:(11)$$H\left(X_{1},X_{2},\cdots ,X_{n}\right)=\sum_{i=1}^{n}H\left(X_{i}|X_{i+1},X_{i+2},\cdots X_{n}\right)$$Based on Equations (9) and (11), one can obtain the following equality:(12)$$\sum_{i=1}^{n}H\left(X_{i}|X_{i-1},X_{i-2},\cdots X_{1}\right)=\sum_{i=1}^{n}H\left(X_{i}|X_{i+1},X_{i+2},\cdots X_{n}\right)$$
which is equivalent to Equation (8). □

## 3. Modeling of Measurement Systems

The unified description and modeling of most measurement systems for all measurement applications is one of the key problems in measurement theory. This paper focuses on the traditional measurement system that provides information about the physical values of measurand [36]. The system has three types of components connected in series, including sensor, variable conversion units, and signal processing units. Sometimes the sensor and variable conversion units are combined.

### 3.1. Model of Measurement Unit

A measurement system [3] consists of a finite number of measurement units as depicted in Figure 2 which is generally a series system. For any unit *i* of the system (i=1,2,⋯,n−1), there are four random variables $X_{i}$, $E_{i}$, $N_{i}$, $X_{i+1}$ associated to it, where $X_{i}$ is input, $E_{i}$ denotes the error, $N_{i}$ is noise (this model only considers additive noise), and with the combined effect of $X_{i}$, $E_{i}$ and $N_{i}$ , the output is $X_{i+1}$. Therefore, the unit can be described by the information entropy-based model in the form of Venn diagram as shown in Figure 1 (for the sake of convenience, here is redrawn as Figure 3) and the entropies of the four variables satisfy:(13)$$H\left(X_{i}\right)+H\left(N_{i}\right)=H\left(X_{i+1}\right)+H\left(E_{i}\right)$$
where
$H\left(X_{i}\right)$ denotes the entropy of the unit input $X_{i}$, $H\left(X_{i+1}\right)$ represents the entropy of output $X_{i+1}$, $H\left(N_{i}\right)=H\left(X_{i+1}|X_{i}\right)$ is the noise entropy that stands for the entropy increase caused by noise, amplification and other reasons, $H\left(E_{i}\right)=H\left(X_{i}|X_{i+1}\right)$ is the error entropy which denotes the information loss of $X_{i}$ passively or proactive, and indicates the active denoising of the measurement unit.

$H\left(X_{i}X_{i+1}\right)$ denotes the joint entropy of $X_{i}$ and $X_{i+1}$, $I\left(X_{i};X_{i+1}\right)$ is the average mutual information between $X_{i}$ and $X_{i+1}$, which denotes the amount of information shared by $X_{i}$ and $X_{i+1}$. The relationships of these entropies satisfy equations as follows:(14)$$\begin{array}{ll} H\left(X_{i}X_{i+1}\right) & =H\left(X_{i}\right)+H\left(X_{i+1}|X_{i}\right)=H\left(X_{i+1}\right)+H\left(X_{i}|X_{i+1}\right)\\ & =H\left(X_{i}|X_{i+1}\right)+I\left(X_{i};X_{i+1}\right)+H\left(X_{i+1}|X_{i}\right) \end{array}$$
(15)$$\begin{array}{ll} I\left(X_{i};X_{i+1}\right) & =H\left(X_{i}\right)-H\left(X_{i}|X_{i+1}\right)=H\left(X_{i+1}\right)-H\left(X_{i+1}|X_{i}\right)\\ & =H\left(X_{i}\right)+H\left(X_{i+1}\right)-H\left(X_{i}X_{i+1}\right). \end{array}$$

The traditional model only considers noise in the signal and the error between the measurement result and true value. In contrast with the traditional model, the proposed model of the measurement unit also considers the information loss in the process of transmission through each measurement unit and can describe the denoising and amplification effect of the measurement unit on the input. These functions of measurement units are represented by error entropy and noise entropy. This shows that this model has excellent ability to describe the measurement unit.

### 3.2. Information Entropy-Based Model of Measurement System

By repeated application of Equation (13), we have the relations of entropies of every unit in a measurement system:(16)$$\left\{ \begin{array}{c} H\left(X_{1}\right)+H\left(N_{1}\right)=H\left(X_{2}\right)+H\left(E_{1}\right)\\ H\left(X_{2}\right)+H\left(N_{2}\right)=H\left(X_{3}\right)+H\left(E_{2}\right)\\ \;\vdots \\ H\left(X_{n-1}\right)+H\left(N_{n-1}\right)=H\left(X_{n}\right)+H\left(E_{n-1}\right). \end{array}\right.$$

Then, adding the two sides of these equations, respectively, and eliminate the same terms, we have:(17)$$H\left(X_{1}\right)+\sum_{i=1}^{n-1}H\left(N_{i}\right)=H\left(X_{n}\right)+\sum_{i=1}^{n-1}H\left(E_{i}\right)$$
where
$H\left(N_{i}\right)=H\left(X_{i+1}|X_{i}\right)$, $H\left(E_{i}\right)=H\left(X_{i}|X_{i+1}\right)$ and Equation (17) is equivalent to:(18)$$H\left(X_{1}\right)+\sum_{i=1}^{n-1}H\left(X_{i+1}|X_{i}\right)=H\left(X_{n}\right)+\sum_{i=2}^{n}H\left(X_{i-1}|X_{i}\right)$$

Equation (18) is the information entropy-based model of measurement system. Notice that Equation (18) is similar to the entropy balance Equation (8). The reason is that the measurement system shown in Figure 2 has a multi-unit serial structure. For the input of the system and outputs of units $X_{1},X_{2},\cdots ,X_{n}$, the random variable $X_{i+1}$ generally only depends on the input $X_{i}$ of the unit i, and is not directly related to the previous random variables $X_{1},X_{2},\cdots ,X_{i-1}$. Therefore, $X_{1},X_{2},\cdots ,X_{n}$ forms a first-order Markov chain, namely:(19)$$H\left(X_{i}|X_{i-1}X_{i-2}\cdots X_{1}\right)=H\left(X_{i}|X_{i-1}\right)$$

Since $X_{1},X_{2},\cdots ,X_{n}$ constitutes a first-order Markov chain, $X_{n},X_{n-1},\cdots ,X_{1}$ is also a first-order Markov chain, that is:(20)$$H\left(X_{i-1}|X_{i}X_{i+1}\cdots X_{n}\right)=H\left(X_{i-1}|X_{i}\right)$$

Therefore, the measurement system can also be described by a first-order Markov chain. Figure 4 depicts the Venn diagram of entropy model of a first-order Markov chain, and this model has a symmetrical structure:

According to the previous discussion, we have:

**Corollary** **1.**
*For a Markov chain*
$X_{n},X_{n-1},\cdots ,X_{1}$
*, the entropy balance equation can be further written as Equation (18).*


**Proof.** According to Theorem 1 and the Markov property, we have Equations (8), (19), and (20). Substituting Equations (19) and (20) into Equation (8) gives Equation (18). □

From Corollary 1, the entropy balance equation of a Markov chain $X_{1}\to X_{2}\to \cdots \to X_{n}$ is the information entropy-based model of measurement system. It shows that all units of a measurement system can be equivalent to one unit as displayed in Figure 5, the sum of all input entropies is equal to the sum of all output entropies.

The information entropy-based model of the measurement system (Equation (18)) not only describes the relationship of the inputs and outputs of the system, but also represents the intermediate quantity in the system, that is, the model of the subsystem can be expressed as: (21)$$H\left(X_{i}\right)+\sum_{k=i}^{j}H\left(X_{k+1}|X_{k}\right)=H\left(X_{j+1}\right)+\sum_{k=i}^{j}H\left(X_{k}|X_{k+1}\right),1\leq i\leq j\leq n-1$$

If the input entropy (or output entropy) and all conditional entropies associated with the subsystem are known, then the subsystem’s output entropy (or input entropy) can be calculated according to Equation (21). 

For an ideal source (the system input is without noise), the measurement result can be directly evaluated by mutual information between the system input and output $I\left(X_{1};X_{n}\right)$. The greater the mutual information, the more accurate the measurement result. The information loss of the measurand can be evaluated by the relative information error (RIE) which is defined as:(22)$$\varepsilon =\frac{H\left(X_{1}\right)-I\left(X_{1};X_{n}\right)}{H\left(X_{1}\right)}\times 100\%$$

An ideal measurement system satisfies $I\left(X_{1};X_{n}\right)=H(X_{1})=H(X_{n})$, and the condition is:(23)$$H\left(X_{1}|X_{n}\right)=H\left(X_{n}|X_{1}\right)=0$$
which means that
$X_{1}$ and $X_{n}$ have the same probability function and information of the measurand is completely acquired by the measurement system.

## 4. Application

To better understand the proposed model, three cases of typical measurement units or processes are discussed in this section.

### 4.1. Case 1: Bandpass Filter

The bandpass filter, which is a typical unit in the measurement system, is analyzed in this section. As shown in Figure 6, the input of the filter K(ω) is Y=X+N, where X is a Gaussian random variable with power of $\sigma_{x}^{2}$, N is white Gaussian noise with power of $\sigma_{n}^{2}$, X and N are independent of each other. The differential entropy of X can be expressed as:(24)$$h\left(X\right)=\frac{1}{2}\log2\pi e\sigma_{x}^{2}$$
and the differential entropy of *N* is denoted by:
(25)$$h\left(N\right)=\frac{1}{2}\log2\pi e\sigma_{n}^{2}$$

Before passing through the filter, since X and N are independent, the power of Y satisfies $\sigma_{y}^{2}=\sigma_{x}^{2}+\sigma_{n}^{2}$. The mutual information between X and Y is:(26)$$\begin{array}{ll} I\left(X;Y\right) & =h\left(Y\right)-h\left(Y|X\right)=h\left(Y\right)-h\left(N\right)\\ & =\frac{1}{2}\log2\pi e\sigma_{y}^{2}-\frac{1}{2}\log2\pi e\sigma_{n}^{2}=\frac{1}{2}\log\left(1+\frac{\sigma_{x}^{2}}{\sigma_{n}^{2}}\right) \end{array}$$

After passing through the filter, the mutual information between X and Z is:(27)$$I\left(X;Z\right)=\frac{1}{2}\log\left(1+\frac{\sigma_{\hat{x}}^{2}}{\sigma_{\hat{n}}^{2}}\right)$$
where
$\sigma_{\hat{x}}^{2}$ and $\sigma_{\hat{n}}^{2}$ represent the power of X and N after pass through the filter, respectively.

The increment of mutual information (IMI) is defined by:(28)$$\Delta I=I\left(X;Z\right)-I\left(X;Y\right)=\frac{1}{2}\log\left(\frac{\sigma_{n}^{2}}{\sigma_{\hat{n}}^{2}}\cdot \frac{\sigma_{\hat{n}}^{2}+\sigma_{\hat{x}}^{2}}{\sigma_{n}^{2}+\sigma_{x}^{2}}\right)$$

Suppose that the power of noise N is $\sigma_{n}^{2}=N_{0}f/2$ where f is the bandwidth of noise and $N_{0}/2$ denotes bilateral power spectral density of noise. The filter is an ideal bandpass filter with a bandwidth of Δf and the gain is 1 in the passband. After passing through the filter, the power of the noise is $\sigma_{\hat{n}}^{2}=N_{0}\Delta f/2$, then Equation (28) can be rewritten as:(29)$$\Delta I=\frac{1}{2}\log\left(\frac{\frac{N_{0}}{2}\cdot f}{\frac{N_{0}}{2}\cdot \Delta f}\cdot \frac{\sigma_{\hat{n}}^{2}+\sigma_{\hat{x}}^{2}}{\sigma_{n}^{2}+\sigma_{x}^{2}}\right)=\frac{1}{2}\log\left(\frac{f}{\Delta f}\cdot \frac{\sigma_{\hat{n}}^{2}+\sigma_{\hat{x}}^{2}}{\sigma_{n}^{2}+\sigma_{x}^{2}}\right)$$

According to the characteristics of the filter, the passband should be consistent with the frequency band of X, that is, $\sigma_{\hat{x}}^{2}=\sigma_{x}^{2}$ and $\sigma_{x}^{2}>>\sigma_{\hat{n}}^{2}$, therefore:(30)$$\Delta I=\frac{1}{2}\log\left(\frac{f}{\Delta f}\cdot \frac{\sigma_{x}^{2}}{\sigma_{n}^{2}+\sigma_{x}^{2}}\right)=\frac{1}{2}\log\left(\frac{f}{\Delta f}\cdot \frac{1}{\frac{\sigma_{n}^{2}}{\sigma_{x}^{2}}+1}\right)$$

Equation (30) shows that the IMI is related to the bandwidth Δf and signal to noise ratio (SNR) of the input signal $\sigma_{x}^{2}/\sigma_{n}^{2}$. The narrower the bandwidth of the filter is, the larger the increment of mutual information is. In general, f/Δf≫1, but the SNR of the input signal $\sigma_{n}^{2}/\sigma_{x}^{2}$ is uncertain. For small signals, the SNR is less than 1 ($\sigma_{x}^{2}/\sigma_{n}^{2}<1$), then we have
(31)$$\Delta I=\frac{1}{2}\log\left(\frac{f}{\Delta f}\cdot \frac{1}{\frac{\sigma_{n}^{2}}{\sigma_{x}^{2}}+1}\right)>0$$

If $\sigma_{x}^{2}/\sigma_{n}^{2}=1$, then:(32)$$\Delta I=\frac{1}{2}\log\left(\frac{f}{2\Delta f}\right)>0$$

For large signal, the SNR is generally much more than 1 ($\sigma_{x}^{2}/\sigma_{n}^{2}\gg 1$), then:(33)$$\Delta I=\frac{1}{2}\log\left(\frac{f}{\Delta f}\right)>0$$

The function of the filter is to filter out the noise contained in the signal. From the above three cases, the IMIs are all greater than zero, which means that at the information level, the role of filter is to increase the amount of information that can be obtained.

### 4.2. Case 2: Quantization Process

The quantization process is an important step in the measurement process. From the perspective of information acquisition, the quantization process is a process of information loss. For a continuous random variable, it requires infinitely high precision to describe itself in theory, and its information entropy is infinite. After quantization, the continuous random variable is transformed into a discrete random variable with limited precision, and its information entropy is finite.

Given a continuous random variable X with a probability density function of p(x), the range of X is evenly divided into intervals of length Δ. Assuming that p(x) is continuous within each interval. According to the mean value theorem, there exists $x_{i}$ within each interval such that:(34)$$p\left(x_{i}\right)\Delta =\int_{i\Delta }^{\left(i+1\right)\Delta }p\left(x\right)dx$$

After quantization, the discrete random variable $X^{Q}$ is obtained and its definition is:(35)$$X^{Q}=x_{i},\;if\;i\Delta \leq X\leq \left(i+1\right)\Delta$$

Then, the probability of $X^{Q}=x_{i}$ is:(36)$$P\left(X^{Q}=x_{i}\right)=\int_{i\Delta }^{\left(i+1\right)\Delta }p\left(x\right)dx=p\left(x_{i}\right)\Delta$$

Therefore, the information entropy of $X^{Q}$ is:(37)$$H\left(X^{Q}\right)=-\sum_{-\infty }^{\infty }P\left(X^{Q}=x_{i}\right)\log P\left(X^{Q}=x_{i}\right)=-\sum p\left(x_{i}\right)\Delta \log p\left(x_{i}\right)-\log\Delta$$

If the function p(x)logp(x) is Riemann integrable, the first item in Equation (37) approaches h(X)=−∫p(x)logp(x)dx as Δ→0, which means:(38)$$\underset{\Delta \to 0}{\lim}H\left(X^{Q}\right)=H\left(X\right)$$

Since Δ→0 is not achievable in practice, there is information loss in the quantization process. For a *N*-bit quantizer, $\Delta =2^{-N}$, then the information loss $H\left(X|X^{Q}\right)$ can be defined as:(39)$$H\left(X|X^{Q}\right)=H\left(X\right)-H\left(X^{Q}\right)\approx -\underset{\Delta \to 0}{\lim}\log\Delta -N\log2$$

The amount of information obtained from *X* with quantization process is:(40)$$I\left(X;X^{Q}\right)=H\left(X^{Q}\right)$$

Therefore, the quantization process can be illustrated as shown in Figure 7. It can be found from Equations (39) and (40) that the larger *N* is, the less information is lost and the more information is obtained.

For example, consider a continuous random variable *X* with uniformly distribution on [0,1]. It is quantized by a *N*-bit quantizer and the process is simulated with MATLAB R2018b (developed by the MathWorks, Inc. with headquarters in Natick, Massachusetts, USA). X is generated by the unifrnd function with 1,000,000 data points. The first 5000 data points of X are shown in Figure 8a, and the probability density function of X is shown in Figure 8b. It can be found that the simulated data of X is not ideal, and its probability density is significantly less than 1 when its value is close to 0 or 1. Here, five quantizers with *N*-bit (N=8,9,10,11,12) are used to quantize X, and then the corresponding information entropies of $X^{Q}$ are calculated and the results are shown in Figure 8c. As h(X)=0, according to Equation (37), the information entropy of $X^{Q}$ is equal to N bits (since $I\left(X;X^{Q}\right)=H\left(X^{Q}\right)$, the mutual information is also N bits), when the log is to the base 2. It can be seen from Figure 8c that the simulation results are consistent with the theoretical values within the allowable error. This also shows that the more bits the quantizer has, the more information can be obtained, which is consistent with the theoretical analysis.

### 4.3. Case 3: Cumulative Averaging Procedure

In some practical measurement applications, the noisy signal is sampled at high speed, then the cumulative averaging procedure is performed to the measured values to filter out the high frequency parts of noise to obtain higher measurement accuracy.

As shown in Figure 9, given a Gaussian signal S with zero mean, a Gaussian noise N with zero mean, S and N are independent of each other and Y=S+N, in a very short period of time Δt, the amplitude of the signal can be considered as constant, and the amplitude of the noise is a variable. Therefore, the correlation coefficient between the signal amplitudes at any two moments in Δt is 1, and for noise, the correlation coefficient is zero. Assuming that the number of cumulative averaging times is *n*, and the power of the signal and noise at each sampling moment $t_{i}$ is $P_{Si}$ and $P_{Ni}$ (i=1,2,⋯,n), then after the cumulative averaging procedure, their power become:(41)$${\bar{P_{S}}}^{*}=\frac{1}{n}\sum_{i=1}^{n}P_{Si}=\frac{1}{n}{\left(A_{S1}+A_{S2}+\cdots +A_{Sn}\right)}^{2}=\frac{1}{n}{\left(n\bar{A_{S}}\right)}^{2}=n{\bar{A_{S}}}^{2}=n\bar{P_{S}}$$
(42)$${\bar{P_{N}}}^{*}=\frac{1}{n}\sum_{i=1}^{n}P_{Ni}=\frac{1}{n}\left(A_{N1}^{2}+A_{N2}^{2}+\cdots +A_{Nn}^{2}\right)=\frac{1}{n}n{\bar{A_{N}}}^{2}={\bar{A_{N}}}^{2}=\bar{P_{N}}$$
where
$A_{Si}$ and $A_{Ni}$ are the amplitudes of the signal and noise at each sampling moment $t_{i}$, respectively; $\bar{A_{S}}$ and $\bar{A_{N}}$ are the average amplitudes of the signal and noise during Δt, respectively; and $\bar{P_{S}}$ and $\bar{P_{N}}$ are the average powers of the signal and noise during Δt, respectively.

After the cumulative averaging procedure, the mutual information that can be obtained from the processed data is:(43)$$I\left(S;Y^{\prime }\right)=\frac{1}{2}\log\left(1+\frac{{\bar{P_{S}}}^{*}}{{\bar{P_{N}}}^{*}}\right)=\frac{1}{2}\log\left(1+\frac{n\bar{P_{S}}}{\bar{P_{N}}}\right)$$
which is greater than the mutual information before the cumulative averaging procedure, that is:(44)$$I\left(S;Y\right)=\frac{1}{2}\log\left(1+\frac{\bar{P_{S}}}{\bar{P_{N}}}\right)$$

This shows that the cumulative averaging procedure can be equivalent to a digital filter, which can improve the signal-to-noise ratio and increase the mutual information. It can also be seen from Equation (43) that the mutual information increases as the number of cumulative averaging times *n* increases.

## 5. Conclusions

In this paper, an information entropy-based modeling method for measurement systems is proposed. The modeling idea of the measurement system based on the viewpoints of information acquisition and uncertainty is presented. Based on this idea, the entropy balance equation based on the chain rule for entropy is proposed for system modeling. Then, information entropy-based models of measurement units and measurement systems are established with the entropy balance equation. Finally, three cases of typical measurement units or processes are analyzed using the proposed model. Compared with the existing modeling methods of measurement systems, the proposed method considers the modeling problem from the perspective of information and uncertainty, and focuses on the loss of the measurand information in the transmission process and the representation of the role of the measurement unit, such as filtering, amplification, and introduced noise. From error entropy, noise entropy, and mutual information between input and output of each unit, the changes of information can be intuitively reflected. If the system input is without noise, the mutual information between the input and output of the system directly reflects the amount of information acquired from measurand, which can be directly used as an evaluation index of the performance of the measurement system.

The proposed model has excellent ability to intuitively describe the processing and changes of information in the measurement system. These characteristics make it easy to have a clear overall understanding of the concept of the measurement system and specific implementation of measurement with measurement units. Note that, although the proposed model has the above advantages, it is not considered and proposed from the perspective of metrological analysis. Compared with the existing models of the measurement system, the output of the proposed model cannot be directly applied to represent the measurement results in the traditional sense, and loses the time information of measurement result. The proposed model does not conflict with the existing models of measurement systems, but can complement the existing models of measurement systems, thus further enriching the existing measurement theory.

## Figures and Tables

**Figure 1 entropy-21-00691-f001:**
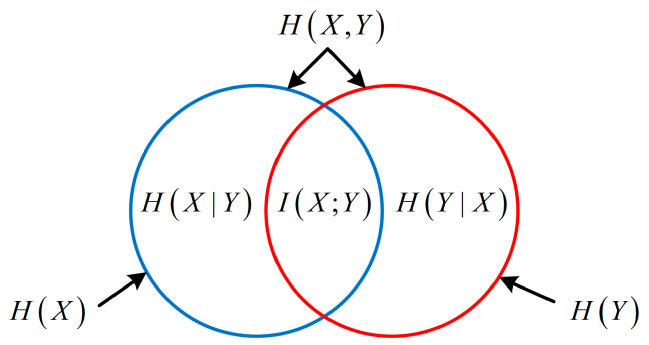
The relationship between various entropies or mutual information.

**Figure 2 entropy-21-00691-f002:**
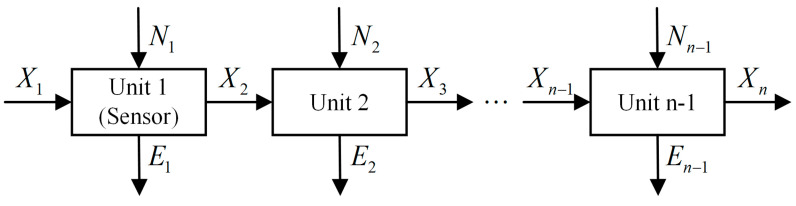
Structure of the actual measurement system.

**Figure 3 entropy-21-00691-f003:**
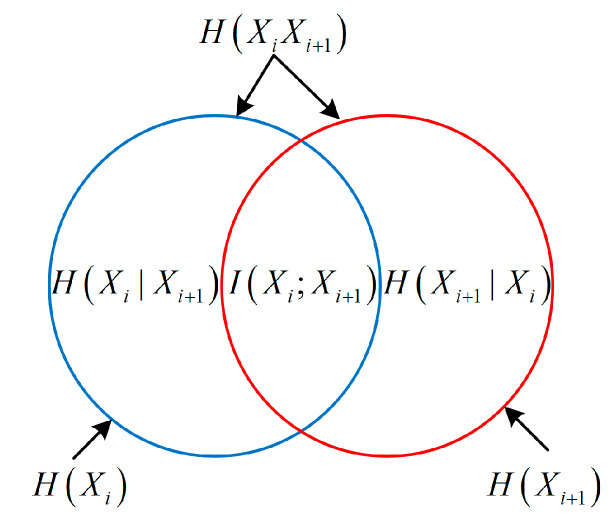
Information entropy-based model of the measurement unit.

**Figure 4 entropy-21-00691-f004:**
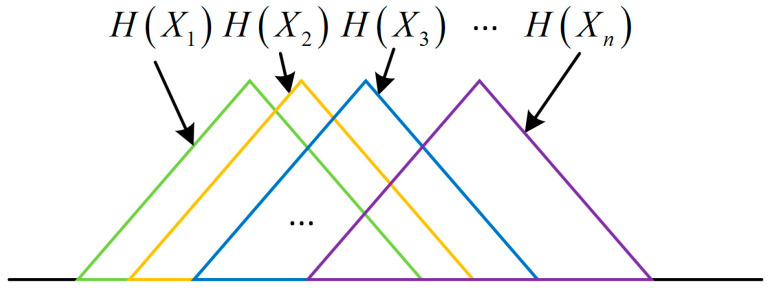
Venn diagram of entropy model of the first-order Markov chain.

**Figure 5 entropy-21-00691-f005:**
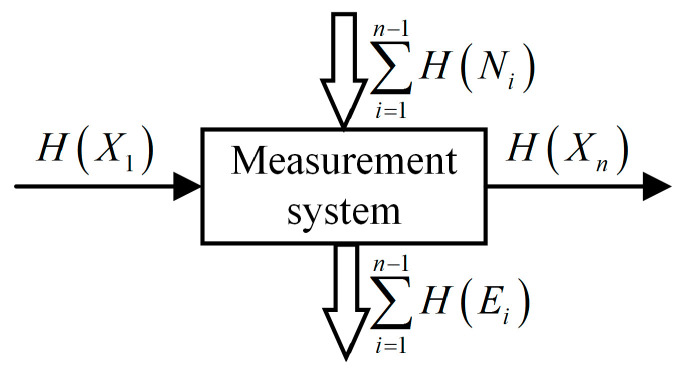
Information entropy-based equivalent model of measurement system.

**Figure 6 entropy-21-00691-f006:**
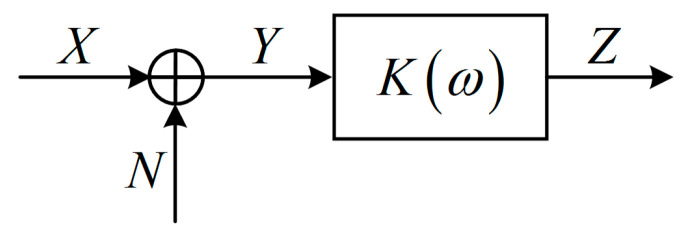
Gaussian random variable with additive white Gaussian noise pass through a bandpass filter.

**Figure 7 entropy-21-00691-f007:**
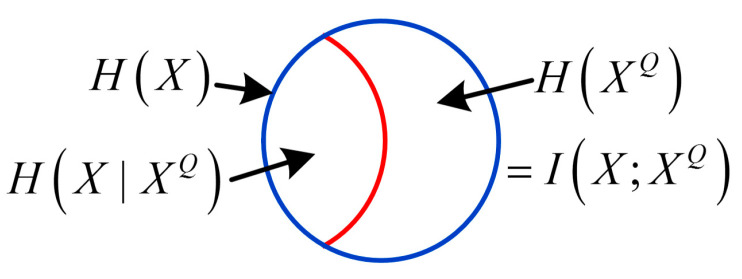
The model of quantization process.

**Figure 8 entropy-21-00691-f008:**
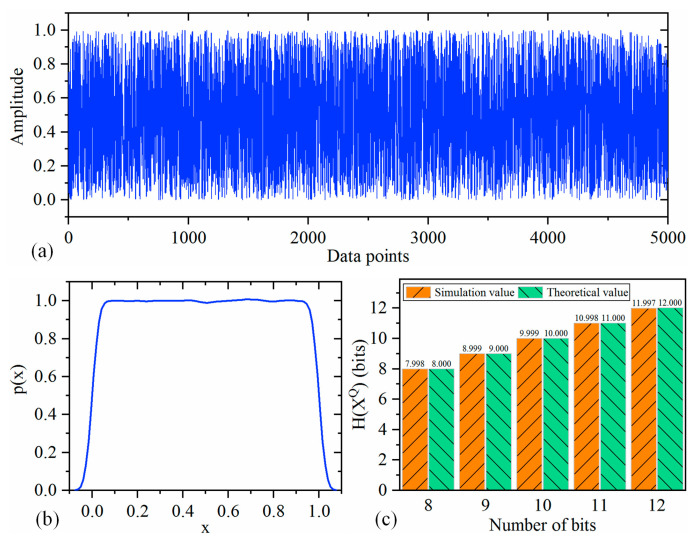
Simulation of quantization process. (**a**) The waveform of the first 5000 data points of the continuous random variable X. (**b**) The probability density function of X. (**c**) Information entropies of $X^{Q}$ quantized by quantizers with different numbers of bits.

**Figure 9 entropy-21-00691-f009:**
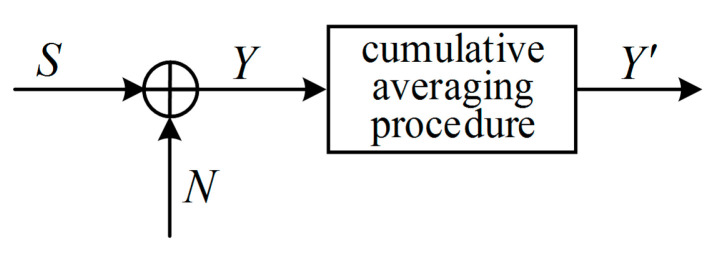
Gaussian random signal with additive Gaussian noise processed by the cumulative averaging procedure.

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
