# Peer review of "An Information Entropy-Based Modeling Method for the Measurement System"

_entropy, 2019, doi:10.3390/e21070691_

Round 1
Reviewer 1 Report
1....H(X), H(Y), H(X|Y), H(X|Y), H(X,Y) and I(X;Y) repeat twice in line 126.
2.The whole paper is very good and can be accepted for publication.
Author Response
Thank you for the comments. We accept the reviewer’s suggestion. Related changes have been marked in red on Page 3 in the marked-up version.
Comment 1. H(X), H(Y), H(X|Y), H(X|Y), H(X,Y) and I(X;Y) repeat twice in line 126.
Response: Thank you for pointing out our mistakes. For two H(X|Y), the latter should be H(Y|X), which we have modified. Please refer to Page 3, Line 130.
Comment 2. The whole paper is very good and can be accepted for publication.
Response: Thank you for your affirmation of this manuscript, we will try our best to improve the quality of the manuscript.
Reviewer 2 Report
Dear Authors,
Please add one or two additional exemplary measurement 'blocks' analysis, similar to the presented bandpass filter. It would be beneficial to understand actual value of the proposed model
Sincerely,
Reviewer
Author Response
Thank you for the comments. We accept the Reviewer’s suggestion. Related changes have been marked in red on Page 9-11 in the marked-up version.
Comment 1. Please add one or two additional exemplary measurement 'blocks' analysis, similar to the presented bandpass filter. It would be beneficial to understand actual value of the proposed model
Response: Thank you for your suggestion. Based on your comments, we have added two examples to the revised manuscript: the quantization process and the cumulative averaging procedure. Please refer to Section 4.2 and 4.3 on Page 9-11.
Related changes have been made in the abstract (Page 1), introduction (Section 1 on Page 2), and conclusion (Section 5 on Page 11).
Reviewer 3 Report
The paper should be improved due to of the below issues:
For what class of the measurement systems the proposed method can be applied?
What is the motivation to choose the bandpass filter as an example of application of the proposed method ?
Why has not the verification of the proposed method for the measurement system defined by the numerical values of related parematers been presented?
Despite the declarations of considering the uncertainty issues, contained in the sections: Abstract and Introduction, they are omitted in the further part of the paper.
Author Response
Thank you for the comments. We accept the reviewer’s suggestion. Related changes have been marked in red on Page 9-11 in the marked-up version.
Comment 1. For what class of the measurement systems the proposed method can be applied?
Response: Thank you for the comment. This paper focuses on the traditional measurement system that provides information about the physical values of measurand. The system has three types of components connected in series, including sensor, variable conversion units, and signal processing units. Sometimes the sensor and variable conversion units are combined. In this paper, the proposed method is used for modeling of the measurement system with series structure. Modeling of measurement systems with other structures by the proposed method is still under consideration.
Comment 2. What is the motivation to choose the bandpass filter as an example of application of the proposed method ?
Response: Thank you for the comment. The proposed method not only describes the entire measurement system, but also the subsystems (even single measurement unit). The reason that bandpass filter is chosen as an application example is that the filter is a representative typical measurement unit in the measurement system, and is a good illustration of the advantages of the proposed method, such as visually characterizing the amount of information obtained and the role of the filter. Based on your comments, we have also added two examples to illustrate the proposed method in the revised manuscript: the quantization process and the cumulative averaging procedure. Please refer to Section 4.2 and 4.3 on Page 9-11.
Comment 3. Why has not the verification of the proposed method for the measurement system defined by the numerical values of related parameters been presented?
Response: Thank you for the comment. The advantage of the proposed method lies in the description, characterization, analysis and evaluation of the measurement system. It is also stated in the text that the proposed method is not considered from the perspective of metrological analysis. Taking this comment into account, we have added an example of the quantization process and corresponding numerical simulation has been conducted to illustrate the proposed method. Please refer to Section 4.2 on Page 9-10.
Comment 4. Despite the declarations of considering the uncertainty issues, contained in the sections: Abstract and Introduction, they are omitted in the further part of the paper.
Response: Thank you for the comment. First, in the introduction, we expounded the basic starting point of considering uncertainty, that is, the measured can be regarded as a random variable rather than the traditionally determined value; secondly, the information entropy itself is a measure of the uncertainty of the random variable, the calculation of which is based on probability statistics. This paper uses the concept of information entropy to model the measurement system, so that the uncertainty is reflected in the full text content.
Reviewer 4 Report
Interesting paper that deals about proposoal of information entropy-based modeling method for measurement system. I have some few comments:
- The research could be supported with some experimental measurement, for example, connection of the proposal as an complement with the existing models of measurement system could be added. Some numerical measurement results could be provided.
- The general idea of the paper is appealing; however, authors should more discussed about practical application of the proposal. The practical significance of this proposal for industrial measurement is not entirely clear. The reader is more interested in practical applications.
- SI units should be added in some variables, e.g. f, N.
I suggest accept this paper but my coments should be taken into account.
Author Response
Thank you for the comments. We accept the reviewer’s suggestion. Related changes have been marked in red on Page 9-11 in the marked-up version.
Comment 1. The research could be supported with some experimental measurement, for example, connection of the proposal as an complement with the existing models of measurement system could be added. Some numerical measurement results could be provided.
Response: Thank you for the comment. We accept the reviewer’s suggestion. In the revised manuscript, we have added the content of the analysis of the quantization process using the proposed method, and numerical simulation has been carried out to illustrate the proposed method. Please refer to Section 4.2 on Page 9-10.
Related changes have been made in the abstract (Page 1), introduction (Section 1 on Page 2), and conclusion (Section 5 on Page 11).
Comment 2. The general idea of the paper is appealing; however, authors should more discussed about practical application of the proposal. The practical significance of this proposal for industrial measurement is not entirely clear. The reader is more interested in practical applications.
Response: Thank you for the comment. The practical application of the proposed method is currently under consideration and research. Based on your valuable suggestions, we use the proposed method to analyze the cumulative averaging techniques commonly used in actual measurements. New content for this has been added in Section 4.3 in the revised manuscript. Please refer to Section 4.3 on Page 10-11.
Related changes have been made in the abstract (Page 1), introduction (Section 1 on Page 2), and conclusion (Section 5 on Page 11).
Comment 3. SI units should be added in some variables, e.g. f, N.
Response: Thank you for the comment which can help further improve the manuscript. According to your comment, we have made changes. A description of the unit of entropy and related measures has been added to the revised manuscript. Please refer to Page 3, Line 102-105.
Round 2
Reviewer 3 Report
I thank the Authors for answering my comments and for improving the paper.